# Comparing cervical cerclage, pessary and vaginal progesterone for prevention of preterm birth in women with a short cervix (SuPPoRT): A multicentre randomised controlled trial

**Natasha L. Hezelgrave[1,2‡], Natalie Suff[1‡*], Paul Seed[1], Vicky Robinson[1], Jenny Carter[1], Helena Watson[1,3], Alexandra Ridout[1], Anna L. David[4], Susana Pereira[5], Fatemeh Hoveyda[6], Joanna Girling[7], Latha Vinayakarao[8], Rachel M. Tribe[1‡], Andrew H. Shennan[1‡], the SuPPoRT Collaborating Team[¶]**

1 Department of Women and Children's Health, School of Life Course and Population Sciences, Faculty of Life Sciences and Medicine, King's College London, London, United Kingdom, 2 Centre for Fetal Care, Queen Charlottes Hospital, Imperial College Healthcare Trust, London, United Kingdom, 3 East Sussex Healthcare NHS Trust, East Sussex, United Kingdom, 4 Elizabeth Garrett Anderson Institute for Women's Health, University College London, London, United Kingdom, 5 The Royal London Hospital, Barts Health NHS Trust, London, United Kingdom, 6 Cambridge University Hospitals NHS Foundation Trust, Cambridge, United Kingdom, 7 West Middlesex University Hospital, Chelsea and Westminster NHS Foundation Trust, London, United Kingdom, 8 Poole Hospital NHS Foundation Trust, Poole, United Kingdom

‡ NLH and NS share first authorship on this work. RMT and AHS are joint senior authors on this work.
¶ Membership of "SuPPoRT Collaborating Team" is provided in the Acknowledgements.
* natalie.suff@kcl.ac.uk

## Abstract

### Background

Cervical cerclage, cervical pessary, and vaginal progesterone have each been shown to reduce preterm birth (PTB) in high-risk women, but to our knowledge, there has been no randomised comparison of the 3 interventions. The SuPPoRT "Stitch, Pessary, or Progesterone Randomised Trial" was designed to compare the rate of PTB <37 weeks between each intervention in women who develop a short cervix in pregnancy.

### Methods and findings

SuPPoRT was a multicentre, open label 3-arm randomised controlled trial designed to demonstrate equivalence (equivalence margin 20%) conducted from 1 July 2015 to 1 July 2021 in 19 obstetric units in the United Kingdom. Asymptomatic women with singleton pregnancies with transvaginal ultrasound cervical lengths measuring <25 mm between $14^{+0}$ and $23^{+6}$ weeks' gestation were eligible for randomisation (1:1:1) to receive either vaginal cervical cerclage ($n = 128$), cervical pessary ($n = 126$), or vaginal progesterone ($n = 132$). Minimisation variables were gestation at recruitment, body mass index (BMI), and risk factor for PTB. The primary outcome was PTB <37 weeks' gestation. Secondary outcomes included

**Data Availability Statement:** All relevant data are within the manuscript and its Supporting information files.

**Funding:** SuPPoRT was funded by National Institute of Health Research (NIHR) fellowship (DRF-2013-06-171 to NH), Tommy's baby charity (Registered charity No. 1060508 to NH, NS, RT, AS), Rosetrees Trust (CM854 to NS, RT), and Borne Foundation (GN2870 to NS, RT). The funders had no role in study design, data collection and analysis, decision to publish, or preparation of the manuscript.

**Competing interests:** AS is a member of PLOS Medicine's editorial board. The authors have declared that no other competing interests exist.

**Abbreviations:** BMI, body mass index; COVID-19, Coronavirus Disease 2019; DMC, Data Monitoring Committee; EPPPIC, Evaluating Progestogens for Preventing Preterm birth International Collaborative; IPD, individual patient data; IUGR, intrauterine growth restriction; NNU, neonatal unit; PPROM, preterm prelabour rupture of fetal membranes; PTB, preterm birth; RCT, randomised controlled trial; RD, risk difference; TSC, Trial Steering Committee; TVUS, transvaginal ultrasound.

PTB <34 weeks', <30 weeks', and adverse perinatal outcome. Analysis was by intention to treat.

A total of 386 pregnant women between $14^{+0}$ and $23^{+6}$ weeks' gestation with a cervical length <25 mm were randomised to one of the 3 interventions. Of these women, 67% were of white ethnicity, 18% black ethnicity, and 7.5% Asian ethnicity. Mean BMI was 25.6. Over 85% of women had prior risk factors for PTB; 39.1% had experienced a spontaneous PTB or midtrimester loss (>14 weeks gestation); and 45.8% had prior cervical surgery. Data from 381 women were available for outcome analysis. Using binary regression, randomised therapies (cerclage versus pessary versus vaginal progesterone) were found to have similar effects on the primary outcome PTB <37 weeks (39/127 versus 38/122 versus 32/132, $p = 0.4$, cerclage versus pessary risk difference (RD) −0.7% [−12.1 to 10.7], cerclage versus progesterone RD 6.2% [−5.0 to 17.0], and progesterone versus pessary RD −6.9% [−17.9 to 4.1]). Similarly, no difference was seen for PTB <34 and 30 weeks, nor adverse perinatal outcome. There were some differences in the mild side effect profile between interventions (vaginal discharge and bleeding) and women randomised to progesterone reported more severe abdominal pain.

A small proportion of women did not receive the intervention as per protocol; however, per-protocol and as-treated analyses showed similar results. The main study limitation was that the trial was underpowered for neonatal outcomes and was stopped early due to the COVID-19 pandemic.

## Conclusions

In this study, we found that for women who develop a short cervix, cerclage, pessary, and vaginal progesterone were equally efficacious at preventing PTB, as judged with a 20% equivalence margin. Commencing with any of the therapies would be reasonable clinical management. These results can be used as a counselling tool for clinicians when managing women with a short cervix.

## Trial registration

EU Clinical Trials register. EudraCT Number: 2015-000456-15, clinicaltrialsregister.eu., ISRCTN Registry: ISRCTN13364447, isrctn.com.

## Author summary

### Why was this study done?

- We evaluated whether commonly used treatments (cervical cerclage, cervical pessary, and vaginal progesterone) are equally effective for preventing preterm birth (PTB) before 37 weeks' gestation in women who are found to have a short cervix on ultrasound scan

**What did the researchers do and find?**

- We randomised 386 at-risk women with a short cervix to one of 3 treatments (cervical cerclage, cervical pessary, and vaginal progesterone).

- Each group had a similar rate of PTB before 37 weeks of gestation.

**What do these findings mean?**

- For women with a short cervix, commencing any of the 3 therapies would be reasonable management for PTB prevention.

- However, as a small number of women received alternative or additional therapies to the randomised treatment, continued monitoring of the cervix is advised.

- Further research should evaluate the benefit of combination therapies.

## Introduction

In clinical practice, a pregnant woman is at higher risk of spontaneous preterm birth (sPTB) if she has a short cervix (<25 mm) identified on transvaginal ultrasound (TVUS) examination in midtrimester (from 14 weeks gestation). Three main prophylactic interventions (cervical cerclage, the cervical "Arabin" pessary, and vaginal progesterone) have shown promise to prevent sPTB for singleton pregnancies, often targeted to those at risk due to a short cervix. Although indirect comparisons of these interventions have been performed [1], the authors are not aware of any single-study direct comparisons to evaluate relative efficacy. The Evaluating Progestogens for Preventing Preterm birth International Collaborative (EPPPIC) meta-analysis of individual patient data (IPD) for vaginal progesterone for prevention of sPTB in women with a short cervix demonstrated a reduction in risk of sPTB <34 weeks of gestation [2]. Similarly, for high-risk women with a short cervix, a meta-analysis of 4 randomised controlled trials (RCTs) [3] revealed that placement of a vaginal cervical cerclage significantly reduced sPTB <35 weeks [4–7]. The pessary was also shown to reduce risk of sPTB <34 weeks of gestation in women with a short cervix [8], although more recent trials and meta-analyses have suggested no benefit [9,10].

Thus, there is no consensus as to the optimal management of high-risk asymptomatic women with a sonographically short cervix. To address this evidence gap, we performed an RCT to directly compare the efficacy of these interventions in singleton pregnancies to prevent preterm birth (PTB) <37 weeks of gestation.

## Methods

### Study design and participants

SuPPoRT was a multicentre, open label 3-arm RCT to compare 3 prophylactic therapies (vaginal cervical cerclage, cervical pessary, and vaginal progesterone) for women with a short cervix in pregnancy. Women were recruited from 19 UK maternity units. The study protocol is published [11]. The study was preapproved by the London-City and East Research Ethics

Committee (reference 15/LO/0485) and authorised by MHRA (28482/0018/001-0001). A Trial Steering Committee (TSC) and a Data Monitoring Committee (DMC) supervised study conduct.

Women were deemed eligible for the study if they had a short cervix (<25 mm) detected on TVUS scan performed by a trained provider according to unit protocol and were between $14^{+0}$ and $23^{+6}$ weeks of gestation with a singleton pregnancy. Women had a cervical length scan because of a risk factor (including previous sPTB or preterm prelabour rupture of fetal membranes (PPROM) <37 weeks of gestation, late miscarriage after 16 weeks of gestation, and history of cervical surgery), but women with an incidental finding of a short cervix on ultrasound were also eligible. Following written informed consent, women were randomised to one of the 3 therapies. Women were excluded if they had persistent vaginal bleeding, visible membranes on speculum, severe abdominal pain, sepsis, PPROM, and contraindications to, or preexisting use of the study interventions. In centres with access to fetal fibronectin testing, a quantitative fetal fibronectin test was carried out prior to initiation of intervention.

### Randomisation

Eligible women were allocated to cerclage, pessary, or progesterone in a 1:1:1 allocation via the MedSciNet web portal (www.medscinet.net). Recruiters and trial coordinators were blind to the randomisation sequence. Minimisation variables were recruitment gestation ($14^{+0}$ to $18^{+6}$ weeks and $19^{+0}$ to $23^{+6}$ weeks), body mass index (BMI, <30 or $\geq$30 kg/m$^2$), and initial risk factor (previous sPTB or late miscarriage, previous cervical surgery, or incidental finding of a short cervix).

### Procedures

Vaginal cervical cerclage: The operative procedure was scheduled within 7 days and was performed by clinicians considered competent in the procedure, according to local practice and guidelines, using their preferred technique and adjunctive therapies (e.g., tocolysis or antibiotics). Cerclage removal was scheduled at 37 weeks of gestation unless the clinical picture mandated earlier removal. Cervical pessary: A suitably sized pessary (as per manufacturer recommendations, guided by parity and presence of a funnel, either $32 \times 65 \times 21$mm or $32 \times 70 \times 21$mm) was inserted within 7 days and removed at 37 weeks of gestation, or earlier if mandated by the clinical picture. If the pessary became dislodged, it was replaced if the participant wished to proceed in the trial. Training in the sizing (and insertion of pessaries was provided to clinicians at all sites. Vaginal progesterone: Participants were prescribed 200 mg of vaginal progesterone once daily (self-administered) from randomisation until 34 weeks of gestation [12]. In all groups, if the cervix shortened after treatment and membranes became visible, participants could be offered an emergency cerclage according to local protocols.

Participants were followed up between 2 and 4 weeks after randomisation, and cervical length was measured by TVUS, as is common practice in the United Kingdom and known to be a good predictor of outcome [13]. Subsequent follow-up appointments were at the discretion of local protocols for PTB prevention. At each follow-up visit, concomitant medication and cervical length measurements were documented, as well as any side effects. A final study visit was carried out at 34 weeks of gestation when unused medication was returned and removal of cerclage or pessary was scheduled (37 weeks).

### Outcomes

The primary outcome was delivery <37 weeks' completed gestation (spontaneous or iatrogenic): Gestational age was calculated according to the first trimester ultrasound dating scan.

Secondary end points (unpowered) were birth <30 and <34 weeks of gestation, gestational age at birth (weeks), and adverse perinatal outcome, defined as a composite outcome of the following: death (midtrimester (14 to 24 weeks) loss, antepartum/intrapartum stillbirths, neonatal deaths prior to discharge from neonatal services), intraventricular haemorrhage, periventricular leukomalacia, hypoxic ischemic encephalopathy, necrotizing enterocolitis, bronchopulmonary dysplasia, and neonatal sepsis (positive blood or central nervous system cultures). Other secondary outcomes included maternal infection at delivery (any of maternal pyrexia >38˚/increased white cell count or c-reactive protein during labour/positive blood cultures/clinical diagnosis of chorioamnionitis/received antibiotics for intrapartum infection), neonatal unit (NNU) at 28 days postdelivery, and intrauterine growth restriction (IUGR) (defined as birth weight <10th centile according to Intergrowth centile charts). Adverse events and reported side effects of treatment (lower abdominal pain, vaginal discharge, vaginal bleeding, vaginal discomfort, difficulty voiding, difficulty with defecation, and any others reported) were also compared between study arms. Other clinical safety outcomes were antenatal complications (antepartum haemorrhage, preeclampsia, gestational diabetes, obstetric cholestasis, and antenatal hospital admissions), onset of labour, mode of delivery, postpartum haemorrhage, postnatal stay, and neonatal outcomes (birthweight, Apgar score, postnatal stay, and requirement for oxygen at 28 days postnatal).

Outcome data were collected by trained midwives, clinicians, and researchers by review of the electronic record and/or handheld patient notes, after discharge from obstetric/neonatal care.

## Sample size calculation

Our previous experience (captured by a robust database of outcome data from >2,000 high-risk women attending our prematurity clinic) indicated that approximately 50% of women with short cervices (<25 mm) treated with cerclage delivered early <37 weeks. From existing published evidence at the time of trial conception, we estimated that vaginal cerclage, cervical pessary, and vaginal progesterone were of approximately equal efficacy to reduce the rate of PTB <37 weeks in high-risk women with a short cervix, with a risk reduction of approximately 50% [8,14,15]. Thus, we estimated that in our population, use of intervention would reduce risk of PTB from approximately 75% (untreated) to around 50%. Equivalence was defined as agreement to within an absolute 20% (e.g., PTB from 60% to 40%). We allowed for differences in both directions in calculating power [16]. Complete data on 170 women per arm (510 in total) were estimated to give 95% power to detect difference clinically important differences of 20% or more in either direction. To allow for possible loss to follow-up (up to 5.5%), the aim was to recruit 540 women.

## Statistical analysis

A statistical analysis plan was finalised before the data were locked and analysis began, according to intention to treat (Stata version 17.1). Baseline demographic, clinical, and procedural data were summarised using descriptive summary statistics, with results reported as numbers (percentages), means (SD), or medians (IQR). The primary and secondary outcomes were compared between treatment groups using binary regression with an identity link for categorical variables, and linear regression for continuous variables, correcting for clustering by study centre using robust standard errors [17]. All pairwise comparisons were considered, and risk differences (RDs) (95% CI) calculated.

An overall test for difference between the 3 arms was performed. As all pairwise comparisons were of interest, a Bonferroni correction was not included. A 2-sided $p$-value of 0.05 was

used to determine statistical significance. Results are reported according to the CONSORT guidelines (Supporting information S1 CONSORT Checklist), considering the extensions for both multiarm and equivalence trials [18,19]. Time between intervention and delivery was analysed using Kaplan–Meier survival analysis. "As-treated" analysis was also conducted for primary and secondary outcomes. Women who received more than 1 intervention were analysed according to the first intervention administered (all women received at least 1 intervention). We also performed a per-protocol analysis (which excluded women exclusion of anyone who did not receive their allocated treatment, or otherwise deviated from the protocol). Data analysis was by intention to treat; however, the proportion of women who did not receive the primary intervention, who stopped the primary intervention early (prior to protocol defined removal), and/or received alternative treatment were also examined.

Three predefined subgroup analyses were performed with interaction tests. Groups were defined by history of previous sPTB/late miscarriage, cervical length <15 mm, and cervicovaginal fetal fibronectin concentration >200 ng/ml preintervention.

## Results

A total of 752 women diagnosed with a short cervix (<25 mm on TVUS) were screened between July 2015 and July 2021. A total of 660 of these satisfied the inclusion criteria, and 386 women (58%) accepted randomisation (Fig 1). This led to 128 women being randomised to cerclage, 126 women to pessary, and 132 women to progesterone. The baseline characteristics of women in each study group were balanced (Table 1). A slightly higher proportion of women in the cerclage group had a cervix <15 mm and a previous midtrimester loss, but these differences were not statistically significant.

Due to an externally imposed halt in recruitment over the Coronavirus Disease 2019 (COVID-19) pandemic, the sample size was reviewed and on the advice of the Independent TSC, and the DMC (who reviewed the unblinded data), the trial was stopped. This was based on having recruited >75% of the intended 510 participants, and that the achieved sample size of 386 gave 85% power to define equivalence as originally planned (agreement within 20%). The data were monitored and locked once all outcomes were collected, and data analysis plan agreed before unblinding.

The number of women randomised per site ranged from 4 to 131. Seven sites did not recruit any women. One woman was randomised to treatment but subsequently withdrew consent for outcome data collection. Five women were lost to follow-up. Obstetric and neonatal outcome data were available for 126 women randomised to cerclage, 125 women randomised to pessary, and 132 women randomised to progesterone. Of the randomised women, 85% (329/385) had risk factors for sPTB and 39% (151/385) had experienced prior PTB/PPROM or late miscarriage. Of those women who were randomised to a cervical cerclage, 94 were performed using the McDonald method, 7 Shirokar (with bladder dissection), and in the remaining women, the technique was not documented ($n = 3$) or a stitch was not placed ($n = 22$). Use of adjuncts at the time of stitch insertion were rare. Indomethacin was used in 7/111 (6.3%) of insertions, and prophylactic antibiotics were given for 25/111 (22.5%) of cases.

The PTB rate of the trial population was 28.3% (108/381). There was no evidence of difference in the primary outcome (rate of PTB <37 weeks) according to allocated treatment; cerclage 29.9% (38/127) versus cervical pessary 31.1% (38/122) versus progesterone 24.2% (32/132) overall difference $p = 0.4$, cerclage versus pessary RD −0.7% (−12.1 to 10.7), cerclage versus progesterone RD 6.2% (−5.0 to 17.0), progesterone versus pessary RD −6.9% (−17.9 to 4.1). Table 2 illustrates RDs ($p$-values) for individual intervention comparisons. Four women had an indicated PTB due to preeclampsia or fetal compromise. There was no difference between

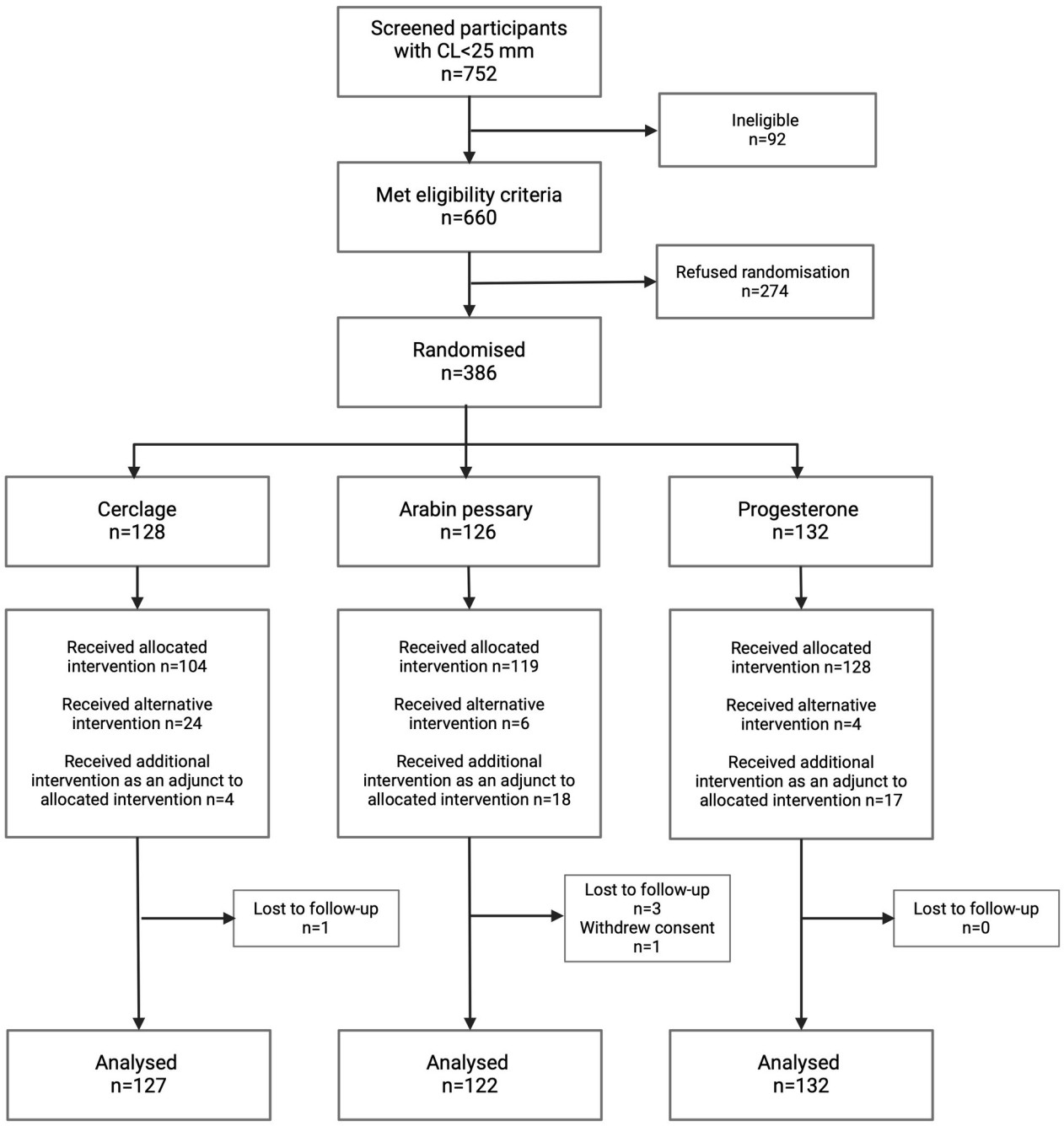

**Fig 1. Trial flow of patients as per CONSORT guidelines.**

intervention arms for composite neonatal outcome, nor PTB at earlier gestations (Table 2). A Kaplan–Meier survival curve of time to delivery showed no differences between intervention arms at any gestation (Fig 2). A similar rate of maternal infection at delivery was noted in the progesterone arm (21.5%) versus 12.7% in the cerclage and 14.0% in the pessary groups (Table 2).

**Table 1. Demographics and baseline characteristics of women randomised in the trial.**

| | Cerclage n = 128 | Pessary n = 125 | Progesterone n = 132 |
|---|---|---|---|
| | % (n) or mean (SD) | % (n) or mean (SD) | % (n) or mean (SD) |
| Age (years) | 32.8 (4.73) | 33.1 (4.73) | 32.9 (4.11) |
| BMI (kg/m$^2$) | 26.0 (5.86) | 25.3 (5.59) | 25.5 (4.80) |
| Ethnicity % (n) | | | |
| - White | 69.5 (89) | 69.6 (87) | 62.1 (82) |
| - Black | 17.2 (22) | 20.8 (26) | 15.9 (21) |
| - Asian | 6.3 (8) | 7.2 (9) | 9.1 (12) |
| - Other | 7.0 (9) | 2.4 (3) | 12.9 (17) |
| Domestic violence (past or present) % (n) | | | |
| - No | 73.2 (93) | 56.0 (70) | 63.6 (84) |
| - Yes | 0.8 (1) | 4.0 (5) | 2.3 (3) |
| - Unknown | 26.0 (33) | 40.0 (50) | 34.1 (45) |
| Past or present history or recreational drug use % (n) | 2.4 (3) | 1.6 (2) | 0.8 (1) |
| Smoking % (n) | | | |
| - Current | 11.0 (14) | 6.4 (8) | 6.1 (8) |
| - Stopped prior to pregnancy | 15.7 (20) | 12.8 (16) | 15.9 (21) |
| - Stopped during pregnancy | 5.5 (7) | 4.0 (5) | 6.8 (9) |
| - Never smoked | 67.7 (86) | 76.8 (96) | 71.2 (94) |
| Any previous pregnancy % (n) | 81.1 (103) | 84.0 (105) | 71.2 (94) |
| Previous pregnancy of at least 14 weeks gestation % (n) | 67.7 (86) | 68.0 (85) | 56.1 (74) |
| Risk factor at booking % (n) | | | |
| - Previous sPTB/PPROM | 30.5 (39) | 34.4 (43) | 28.0 (37) |
| - Previous midtrimester loss | 15.6 (20) | 11.2 (14) | 13.6 (18) |
| - Previous cervical surgery | 55.5 (71) | 53.6 (67) | 57.6 (76) |
| - Incidental finding of short cervix | 15.6 (20) | 15.2 (19) | 12.9 (17) |
| Previous sPTB /PPROM/midtrimester loss % (n) | 41.7 (53) | 38.4 (48) | 37.9 (50) |
| Previous cervical surgery only % (n) | 42.2 (54) | 46.4 (58) | 49.2 (65) |
| Cervical length % (n) | | | |
| 15–24 mm | 87.5 (112) | 88.0 (110) | 91.7 (121) |
| <15 mm | 12.5 (16) | 12.3 (15) | 8.3 (11) |
| Mean gestation of intervention, weeks[+d] (SD)/n[a] | 20[+1] (2.7)/104[a] | 19[+5] (2.5) /119[a] | 19[+6] (2.6) /128[a] |

[a]Different overall sample size due to missing data or patients receiving alternative.

BMI, body mass index; PPROM, premature prelabour rupture of membranes; SD, standard deviation; sPTB, spontaneous preterm birth.

Emergency sutures for bulging fetal membranes were inserted for 14 women (9 women randomised to progesterone and 5 randomised to cervical pessary). There was a significantly higher rate of rescue cerclage insertion in the pessary and progesterone groups, compared with cerclage group (4.1% and 7.7%, respectively, versus 0%; $p < 0.001$).

No difference was seen in primary outcome when predefined subanalysis was performed (S2 Table), although numbers were small, so caution in interpretation is advised. There were no obvious differences in the predefined safety outcomes between treatment arms (S3 Table).

Once randomised, a small proportion of women did not receive the intervention according to protocol (either an alternative or additional intervention was used) (Fig 1). A total of 24 women randomised to cerclage did not have it placed due to either clinician inability to place the cerclage at time of insertion ($n = 5$) or patient request ($n = 19$). Of these, they all received

**Table 2. Primary and secondary outcomes (and their components) according to randomised intervention.**

| | Cerclage % (n/ 127) | Pessary % (n/ 122) | Progest % (n/ 132) | P value for overall difference | Cerclage vs Pessary RD % (CI) | Cerclage vs Progest RD % (CI) | Progest vs Pessary RD % (CI) |
|---|---|---|---|---|---|---|---|
| PTB <37 weeks of gestation | 30.5 (39) | 31.2 (38) | 24.2 (32) | 0.4 | −0.7 (−12.1 to 10.7) | 6.2 (−5.0 to 17.0) | −6.9 (−17.9 to 4.1) |
| Adverse perinatal outcome | 8.6 (11) | 7.2 (9) | 12.1 (16) | 0.4 | 1.4 (−5.2 to 8.0) | −3.5 (−10.9 to 3.9) | 4.9 (−2.3 to 12.1) |
| Components of the adverse perinatal outcome | | | | | | | |
| - Midtrimester loss/Stillbirth/Perinatal death | 5.5 (7) | 5.6 (7) | 7.6 (10) | 0.7 | −0.1(−5.8 to 5.5) | −2.1 (−8.1 to 3.9) | 2.0 (−4.1 to 8.0) |
| - PVL | 0 | 0 | 0 | - | - | - | - |
| - HIE | 0 | 0 | 1.5 (2) | 0.2 | - | −1.5 (−3.6 to 0.6) | 1.5 (−0.6 to 3.6) |
| - NEC | 0 | 1.6 (2) | 0.8 (1) | 0.3 | −1.6 (−3.9 to 0.6) | −0.8 (−2.2 to 0.7) | −0.9 (−3.6 to 1.8) |
| - BPD | 2.4 (3) | 1.6 (2) | 2.3 (3) | 0.9 | 0.7 (−2.7 to 4.2) | 0.02 (−3.7 to 3.7) | 0.7 (−2.8 to 4.2) |
| - Neonatal sepsis | 0.8 (1) | 1.6 (2) | 2.3 (3) | 0.6 | −0.9 (−3.6 to 1.9) | −1.6 (−4.6 to 1.5) | 0.7 (−2.8 to 4.2) |
| - IVH | 0.8 (1) | 0.8 (1) | 0.8 (1) | 1.0 | −0.04 (−2.2 to 2.2) | 0.02 (−2.1 to 2.1) | −0.1(−2.2 to 2.1) |
| PTB <34 weeks | 18.1 (23) | 18.0 (22) | 16.7 (22) | 0.9 | 0.08 (−9.4 to 9.6) | 1.4 (−7.8 to 10.6) | −1.4 (−10.7 to 8.0) |
| PTB <30 weeks | 10.2 (13) | 8.2 (10) | 11.4 (15) | 0.7 | 2.0 (−5.1 to 9.2) | −1.1 (−8.6 to 6.4) | 3.2 (−4.1 to 10.4) |
| Time between intervention and delivery Days mean (SD) | 114.6 (±35.8) | 116.6 (±33.8) | 119.0 (±38.1) | 0.6 | −199.4 (−1,149 to 750) | −439.0 (−1,360 to 483) | 239.3 (−662 to 1,141) |
| Maternal infection | 12.7 (16/126[a]) | 14.0 (17/121[a]) | 21.5 (28/130[a]) | 0.1 | −1.4 (−9.8 to 7.1) | −8.8 (−18.0 to 0.3) | 7.5 (−1.9 to 16.9) |
| IUGR (<10th Intergrowth centile) | 3.3 (4/121[a]) | 6.7 (8/119[a]) | 3.3 (4/121[a]) | 0.3 | −3.4 (−8.9 to 2.1) | 0 (−4.5 to 4.5) | −3.4 (−8.9 to 2.1) |
| Baby in NNU at 28 days | 2.4 (3) | 6.6 (8) | 5.3 (7) | 0.3 | −4.2 (−9.3 to 0.9) | −3.0 (−7.6 to 1.7) | −1.3 (−7.0 to 4.6) |
| Neonatal sepsis (by blood culture) | 0.8 (1/127[a]) | 1.6 (2/122[a]) | 2.3 (3/128[a]) | 0.6 | −0.01 (−3.5 to 1.8) | −1.6 (−4.6 to 1.5) | 0.7 (−2.8 to 4.2) |

[a]Different overall sample size due to missing data.

BPD, bronchopulmonary dysplasia; HIE, hypoxic ischaemic encephalopathy; IUGR, intrauterine growth restriction; IVH, intraventricular haemorrhage; NEC, necrotising enterocolitis; NNU, neonatal unit; PTB, preterm birth; PVL, periventricular leukomalacia; RD, risk difference. Vaginal Progesterone abbreviated to Progest.

progesterone as an alternative treatment. A further 4 women were prescribed progesterone after cerclage insertion (3 on clinician advice, 1 patient request). In the pessary group, 2 patients were assigned alternative treatment after the pessary became dislodged twice, 8 due to patient request (after vaginal bleeding or discomfort or continued cervical shortening), and 4 after clinician advice due to continued cervical shortening. Six women requested alternative treatment prior to insertion. In the progesterone group, 11 women received a cerclage, 5 at clinician request after continued cervical shortening, and 4 as patient request after randomisation. However, when the "as-treated" analysis was undertaken (according to primary treatment received), similar results for primary and secondary outcomes were obtained (Table 3 and S1 Fig). A per-protocol analysis was also performed, which showed similar results (S4 Table).

S5 Table illustrates the side effect profile of the 3 treatment arms. Women randomised to progesterone reported significantly more severe lower abdominal pain than those with

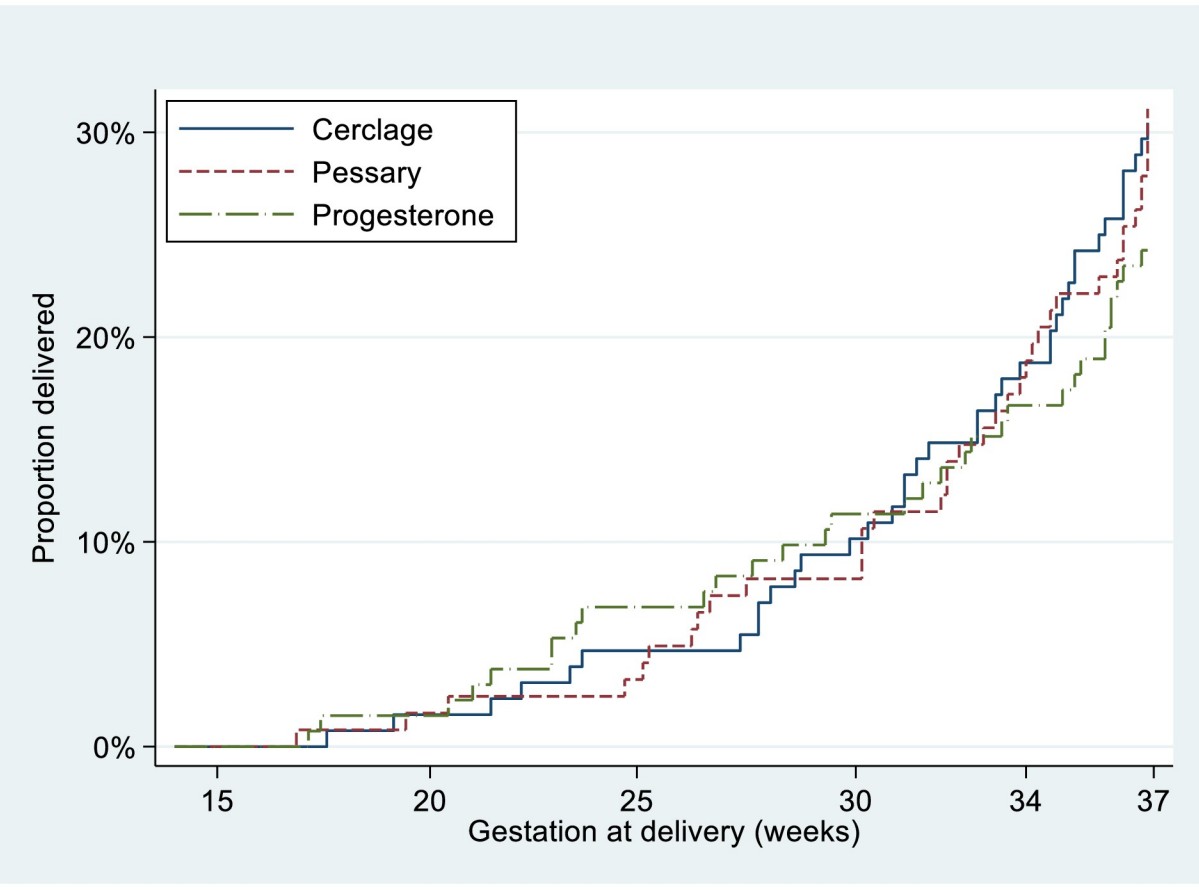

**Fig 2. Kaplan–Meier survival analysis for 3 treatment arms (intention-to-treat analysis).**

cerclage and pessary, although numbers were small, and incidence of mild abdominal pain was similar across groups. Mild vaginal bleeding was more frequent in the pessary group. Women in the pessary and progesterone trial arms were more likely to report mild and severe increase in vaginal discharge compared with the cerclage group.

## Discussion

SuPPoRT is, to our knowledge, the only randomised trial directly comparing the 3 common therapies for women with a short cervix. In this pragmatic trial, we show that there is no clear advantage between the 3 therapies to prevent PTB before 37 weeks' gestation, as judged with a 20% equivalence margin. Commencing any of these therapies as first-line treatment of a short cervix would be reasonable, within a shared decision-making model. Results remained consistent when actual treatment received was analysed, although given these were driven by patient or clinical request in a small proportion of women, there is potential for treatment bias and must thus be interpreted with caution. There were also no significant differences in secondary outcomes between the groups, although maternal infection, often associated with PTB, may be higher in the progesterone group. While this did not meet statistical significance, it is possible that repeated digital insertion of progesterone pessary may contribute towards higher rates of maternal infection during labour. However, histological chorioamnionitis was similar between the 3 groups. Nevertheless, this finding warrants additional investigation.

**Table 3. Primary and secondary outcomes (and their components) for women in the SuPPoRT study according to as-treated analysis.**

| | Cerclage % (n/111) | Pessary % (n/116) | Progest % (n/155) | P value for overall difference | Cerclage vs. Pessary RD % (CI) | Cerclage vs. Progest RD % (CI) | Progest vs. Pessary RD % (CI) |
|---|---|---|---|---|---|---|---|
| PTB <37 weeks of gestation | 30.6 (34) | 32.8 (38) | 23.9 (37) | 0.2 | −2.1 (−14.2 to 10.0) | 6.8 (−4.1 to 17.6) | −8.9 (−19.7 to 2.0) |
| Adverse perinatal outcome | 9.0 (10) | 7.6 (9) | 11.0 (17) | 0.6 | −1.4 (−5.7 to 8.6) | 2.8 (−6.5 to 12.1) | 3.4 (−3.4 to 10.2) |
| PTB <34 weeks | 18.9 (21) | 19.0 (22) | 16.1 (25) | 0.8 | −0.05 (−10.2 to 10.2) | −3.3 (−5.9 to 12.6) | −2.8 (−12.0 to 6.4) |
| PTB <30 weeks | 10.8 (12) | 8.6 (10) | 10.3 (16) | 0.8 | 2.2 (−5.5 to 9.9) | 0.5 (−7.0 to 8.0) | 1.7 (−5.3 to 8.7) |
| Time between intervention and delivery days mean (SD) | 114.6 (± 35.7) | 116.6 (± 33.8) | 119.1 (±38.2) | 0.6 | −205 (−1,155 to 745) | −449 (−1,371 to 473) | 244 (−658 to 1,146) |
| Maternal Infection | 11.9 (13/ 109[a]) | 13.9 (16/ 115[a]) | 20.9 (32/ 153[a]) | 0.1 | −2.0 (−10.8 to 6.8) | −9.0 (−17.9 to −1.3) | 7.0 (−2.0 to 16.0) |
| IUGR (<10th Intergrowth centile) | 1.9 (2/104[a]) | 7.1 (8/113[a]) | 4.2 (6/144[a]) | 0.2 | −5.2 (−10.6 to 0.3) | −2.2 (−6.4 to 2.0) | −2.9 (−8.7 to 2.8) |
| Baby in NNU at 28 days | 2.7 (3) | 6.9 (8) | 4.4 (7) | 0.3 | −4.1 (−9.7 to 1.3) | −1.8 (−6.3 to 2.6) | −2.4 (−8.0 to 3.3) |
| Neonatal sepsis (by blood culture) | 0.9 (1/110[a]) | 1.7 (2/116[a]) | 2.0 (3/151[a]) | 0.8 | −0.8 (−3.8 to 2.1) | −1.1 (−3.9 to 1.8) | 0.3 (−3.0 to 3.5) |

[a]Different overall sample size due to missing data.

BPD, bronchopulmonary dysplasia; HIE, hypoxic ischaemic encephalopathy; IUGR, intrauterine growth restriction; IVH, intraventricular haemorrhage; NEC, necrotising enterocolitis; NNU, neonatal unit; PTB, preterm birth; PVL, periventricular leukomalacia; RD, risk difference. Vaginal Progesterone abbreviated to Progest.

One of the main study limitations was that a small number of women received alternative or additional therapies to that to which they were randomised. Introduction of additional therapies postrandomisation were largely due to clinician or patient anxiety. In the context of regular postinterventional monitoring as part of the trial, in some women, continuing shortening of the cervix led to emergency cervical cerclage for bulging fetal membranes. As 4% and 7% of women, respectively, in the pessary and progesterone groups received a rescue cerclage, this suggests that it may be beneficial to continue postinterventional TVUS monitoring in these women. Furthermore, there were several women in whom cerclage placement was difficult due to a particularly deficient cervix and/or operative experience of the surgeon, and so other therapies, namely progesterone, were alternatively initiated. Some women allocated to cerclage requested different management. Despite this, the as-treated analyses did not show any effect from protocol deviation, suggesting that our overall study results are still valid. However, some clinicians, as well as women, are likely to favour noninvasive, operator-independent therapies, and some women (as well as clinicians) favoured operative intervention (cerclage) despite randomisation to other therapy. Reassuringly, side effects were similar between the groups, although vaginal discharge was more often reported by women using the pessary and progesterone. Evaluation of the benefit of combination therapies in future randomised controlled trials is indicated.

A further limitation was that the trial was stopped early due to the COVID-19 pandemic, although adequate power was achieved to evaluate the primary outcome as judged by the DMC. Finally, the study was underpowered for the arguably clinically more significant outcome of PTB prior to 30 and 34 weeks of gestation. Women with a short cervix are relatively uncommon, so the numbers achieved in this study are substantial (386 randomised). Even in a large UK progesterone trial, which included 66 sites, only 256 women with a short cervix were randomised [20]. Moreover, randomisation was challenging when there are strongly held

beliefs about the perceived risk/benefit from patients and clinicians. Many women who had experienced pregnancy loss or PTB had strong preferences for particular interventions and were unwilling to be randomly assigned. Thus, recruitment took longer than anticipated, despite number of centres. Numbers of women with very short cervixes <15 mm were also too low to confidently assess treatment effect for the highest-risk women.

A recent survey of UK PTB prevention clinical practice found that a wide variety of treatment regimens and treatment combinations are currently offered; only 19% of UK PTB clinics currently use vaginal progesterone as first-line treatment for a short cervix, and none offer the pessary routinely to women at high risk [21]. This is likely due to uncertainties regarding the comparative benefit and at the time of trial commencement, evidence for progesterone treatment for all women with a history of PTB was lacking. While there have been many published trials comparing 1 preventative intervention with placebo, only 1 other small feasibility study with limited power (18 participants) has directly compared the 3 commonly used interventions [22]. We have previously compared progesterone (n = 19) with cerclage (n = 17) in a small RCT to gain insight into their mechanism of action. Mean gestational age at birth was similar for each intervention; however, only cerclage, and not progesterone, altered the inflammatory environment and positively affected cervical length [23].

Large RCTs assessing progesterone therapy for PTB prevention had cast doubt on the treatment effectiveness [20]; however, results from the more recent EPPPIC IPD meta-analysis have supported the use of vaginal progesterone for prevention of PTB <34 weeks' gestation in women with a short cervix in singleton pregnancy [2], and a network meta-analysis (61 trials of PTB interventions) supported the use of progesterone and potentially the use of cerclage to prevent PTB in women at high risk [1]. The cervical pessary, however, showed limited efficacy, and a recent RCT (544 women with cervical length <20 mm) has suggested that it may be associated with fetal or neonatal harm [24]. Thus, in the absence of more reassuring data, cerclage and progesterone may be the preferred interventions of choice. Most of the published trials included women with a short cervix, a history of sPTB, or both, as these groups often overlap in clinical practice. Our study, however, only included women with a short cervix; some of these women would have had a history of sPTB, but the majority did not (>60%). Therefore, comparison with previous trials is difficult. Given the breadth of risk factors for sPTB included in this pragmatic trial, it is possible that pooling patients limited the impact of treatments or masked differences between them, although subgroup analysis was reassuring. As we learn more about the heterogeneous mechanisms of sPTB, future studies should explore subgroups of women at high risk. Given the recruitment challenges encountered by large individualised RCTs such as this one, novel research methodologies may be required.

## Conclusions

In this large RCT, there was no clear advantage in PTB prevention between the 3 therapies studied. Further research is needed into the benefits of combination and sequential therapies, and whether the underlying aetiology of PTB may influence choice and efficacy of prophylactic treatment.

## Supporting information

**S1 Table. Variables used for minimisation in the trial.** BMI, body mass index; sPTB, spontaneous preterm birth. Vaginal progesterone abbreviated to Progest.
(DOCX)

**S2 Table. Prespecified subgroup analyses based on baseline risk factors of randomised women.** CL, cervical length; fFN, fetal fibronectin; PTB, preterm birth; sPTB, spontaneous preterm birth. Vaginal Progsterone abbreviated to Progest.
(DOCX)

**S3 Table. Safety outcomes for intention-to-treat analysis.** APH, antepartum haemorrhage; CS, cesarean section; GDM, gestational diabetes mellitus; ICP, intrahepatic cholestasis of pregnancy; PPH, postpartum haemorrhage; SVD, spontaneous vaginal delivery.
(DOCX)

**S4 Table. Primary and secondary outcomes (and their components) for women in the SuP-PoRT study according to per-protocol analysis.**
(DOCX)

**S5 Table. Reported side effect profiles for women randomised to cerclage, Arabin pessary, and vaginal progesterone.**
(DOCX)

**S1 Fig. Kaplan–Meier survival analysis for 3 treatment arms (as-treated analysis).**
(TIF)

**S1 CONSORT checklist. CONSORT 2010 checklist of information to include when reporting a randomised trial.**
(DOC)

**S1 CONSERVE checklist. Use CONSERVE-CONSORT for completed trial reports and CONSERVE-SPIRIT for trial protocols.**
(DOCX)

**S1 Statistical Analysis Plan. Statistical analysis plan for SuPPoRT trial.**
(DOCX)

**S1 Trial Protocols. SuPPoRT trial study protocol documents.**
(ZIP)

## Acknowledgments

We thank the members of the Trial Steering (Jason Waugh, Melanie Griffin, Su Harper-Clark) and Data Monitoring Committee (Mark Johnson, Brenda Kelly, Vichithranie Madurasinghe) and all the people who helped in the conduct of the study from all the recruiting centres (St Thomas' Hospital NHS Trust, Cambridge University Hospital, West Middlesex University Hospital, Liverpool Women's NHS Foundation Trust, Poole Hospital NHS Foundation Trust, Kingston Hospital NHS Foundation Trust, University College Hospital London, Leicester Royal infirmary, Norfolk and Norwich University Hospital NHS Trust, South Tyneside and Sunderland NHS Trust). We specifically thank the members of the 'SuPPoRT Collaborating Team' (Manju Chandiramani, Lisa Story, Evonne Chin Smith, Cally Gill, Amirah Mohd Zaki, Naomi Carlisle, Debbie Finucane, Stephanie Grigsby, Elena Romero, Sarah Weist, Kirsten Herdman, Lesley Hewitt, Eileen Walton, Amy Barker, Laura Harris, Kristina Sexton, Isabel Bradley, Siobhan Holt, Michelle Dower, Borna Poljak, Devender Roberts, Sarah Weist, Zarko Afirevic, Angharad Care, Laura Goodfellow, Melissa Whitworth, Nigel Simpson, Penelope McParland, Alexandra Patience, John Lartey, Kim Hinshaw, and the NIHR CRN team). We are grateful to Birgit Arabin who provided the cervical pessaries free of charge. We are also

grateful to the many people who helped in this study but who we have been unable to name, and in particular all the women (and their babies) who participated in the trial.

## Author Contributions

**Conceptualization:** Natasha L. Hezelgrave, Natalie Suff, Jenny Carter, Joanna Girling, Latha Vinayakarao, Rachel M. Tribe, Andrew H. Shennan.

**Data curation:** Natasha L. Hezelgrave, Natalie Suff, Paul Seed, Jenny Carter, Helena Watson, Anna L. David, Susana Pereira, Fatemeh Hoveyda, Joanna Girling, Latha Vinayakarao, Rachel M. Tribe, Andrew H. Shennan.

**Formal analysis:** Natasha L. Hezelgrave, Natalie Suff, Paul Seed, Jenny Carter, Rachel M. Tribe, Andrew H. Shennan.

**Funding acquisition:** Natasha L. Hezelgrave, Natalie Suff, Rachel M. Tribe, Andrew H. Shennan.

**Investigation:** Natasha L. Hezelgrave, Natalie Suff, Jenny Carter, Anna L. David, Fatemeh Hoveyda, Joanna Girling, Latha Vinayakarao, Rachel M. Tribe, Andrew H. Shennan.

**Methodology:** Natasha L. Hezelgrave, Paul Seed, Jenny Carter, Rachel M. Tribe, Andrew H. Shennan.

**Project administration:** Natasha L. Hezelgrave, Natalie Suff, Vicky Robinson, Jenny Carter, Helena Watson, Alexandra Ridout, Anna L. David, Susana Pereira, Fatemeh Hoveyda, Joanna Girling, Latha Vinayakarao, Rachel M. Tribe.

**Resources:** Susana Pereira, Joanna Girling.

**Software:** Vicky Robinson, Jenny Carter, Alexandra Ridout.

**Validation:** Natalie Suff, Paul Seed.

**Visualization:** Natalie Suff, Jenny Carter, Rachel M. Tribe.

**Writing – original draft:** Natasha L. Hezelgrave, Natalie Suff, Andrew H. Shennan.

**Writing – review & editing:** Natasha L. Hezelgrave, Natalie Suff, Paul Seed, Vicky Robinson, Jenny Carter, Helena Watson, Alexandra Ridout, Anna L. David, Susana Pereira, Fatemeh Hoveyda, Joanna Girling, Latha Vinayakarao, Rachel M. Tribe, Andrew H. Shennan.

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
