## [Editor Report · Decision Letter 0]

22 Dec 2023

Dear Dr Suff, 

Thank you for submitting your manuscript entitled "SuPPoRT: a multi-centre randomised controlled trial comparing cervical cerclage, pessary and vaginal progesterone, for the prevention of preterm birth in women with a short cervix" for consideration by PLOS Medicine.

Your manuscript has now been evaluated by the PLOS Medicine editorial staff and I am writing to let you know that we would like to send your submission out for external assessment.

However, we first need you to complete your submission by providing the metadata that are required for full assessment. To this end, please login to Editorial Manager where you will find the paper in the 'Submissions Needing Revisions' folder on your homepage. Please click 'Revise Submission' from the Action Links and complete all additional questions in the submission questionnaire.

Please also:

Correct "1 July 2025" in the abstract; and

Include the full trial protocol and, if available, statistical analysis plan as supplementary document(s). 

Please re-submit your manuscript by the first week of January.

Once your full submission is complete, your paper will undergo a series of checks in preparation for full assessment. 

Sincerely,

Richard Turner PhD

Consulting Editor, PLOS Medicine

plosmedicine@plos.org

---

## [Decision Letter · Decision Letter 1]

7 Feb 2024

Dear Dr. Suff,

Thank you very much for submitting your manuscript "SuPPoRT: a multi-centre randomised controlled trial comparing cervical cerclage, pessary and vaginal progesterone, for the prevention of preterm birth in women with a short cervix" (PMEDICINE-D-23-03700R1) for consideration at PLOS Medicine. 

Your paper was discussed with an academic editor with relevant expertise and sent to independent reviewers, including a statistical reviewer. The reviews are appended at the bottom of this email and any accompanying reviewer attachments can be seen via the link below:

[LINK]

In light of these reviews, we will not be able to accept the manuscript for publication in the journal in its current form, but we would like to invite you to submit a revised version that addresses the reviewers' and editors' comments fully. You will appreciate that we cannot make a decision about publication until we have seen the revised manuscript and your response, and we expect to seek re-review by one or more of the reviewers. 

We hope to receive your revised manuscript by Mar 04 2024 11:59PM. Please email us (plosmedicine@plos.org) if you have any questions or concerns.

Please let me know if you have any questions, and we look forward to receiving your revised manuscript. 

Sincerely,

Richard Turner PhD

Consulting editor, PLOS Medicine

plosmedicine@plos.org

Please adapt the title to PLOS Medicine style, and we suggest: "Comparing cervical cerclage, pessary and 

vaginal progesterone for prevention of preterm birth in women with a short cervix (SuPPoRT): A multi-centre randomised controlled trial". 

We ask you to adapt the abstract to the three-part journal style. The final sentence of the 'Methods and findings' subsection should begin "Study limitations include ..." or similar and should quote 2-3 of the study's main limitations.

Please mention stopping the trial in the abstract. 

Please substitute the 'Key points' with an 'Author summary' section in journal style (you may find it helpful to consult one or two recent PLOS Medicine research papers to get a sense of the preferred style). 

Rather than "In women with a short cervix, cerclage ... are equally efficacious ..." (as in the current abstract), we ask you to adapt the conclusions throughout to "In this study, we found that ... were equally efficacious ... as judged with a 20% equivalence margin." or similar. 

Please ensure that the Discussion section (main text) includes a paragraph on study limitations. 

Please include a completed CONSORT checklist with your revision, labelled 'S1_CONSORT_Checklist' or similar and referred to as such in the text. 

In the checklist, please refer to individual items by section (e.g., "Methods") and paragraph number, not by line or page numbers as these generally change in the event of publication. 

Please also consider using the CONSERVE extension if appropriate. 

Comments from the reviewers:

*** Reviewer #1: 

Alex McConnachie, Statistical Review

Hezelgrave et al present the results of a three-arm equivalence trial of treatments for the prevention of preterm birth in women with a short cervix. This review considers the statistical aspects of their paper.

I found this to be a well written and interesting paper. The very high follow-up rate is a strength, and to be commended. I have a number of comments about the analysis, though these should all be possible to address.

The sample size calculation is presented as it is described in the protocol, which was reviewed by the funders and approved by the regulators, and in that sense is correct. The fact that it is based on a two-group test of superiority, rather than equivalence, is unfortunate, though in fact makes little difference. By assuming that the average primary outcome rate to be 50%, the sample size calculation was actually conservative, and the sample size required would have been smaller for any other true average (as turned out to be the case).

Quite rightly for an equivalence study, the authors say that they consider differences in treatment effect in either direction, citing Jones (1996). However, Jones (1996) is quite clear in that the analysis should be based on confidence intervals. For any two treatments, equivalence can be declared if the confidence interval for the treatment effect difference lies entirely within the equivalence limits, in this case the range from -20% to +20%. The approach taken in this paper, of declaring equivalence if the analysis comparing all three treatments is not statistically significant at 5%, is not the same thing. For example, a very small study would be unlikely to show statistical significance between groups, yet the confidence intervals for differences between treatments would be wide, and likely not be contained within the equivalence limits. On the other hand, a very large study might be able to detect treatment effect differences, even if these differences are small, and the treatments should be declared equivalent within the specified limits.

I suggest that the overall test of differences between groups has no bearing on the equivalence or otherwise of the three treatments, and should not be the main focus of the analysis. The things that matter are the estimated differences between treatments, and their confidence intervals. If these intervals lie within the equivalence limits, then equivalence can be declared. According to Tables 2 and 3 in the paper, this appears to be the case (just), so it seems that the overall conclusion of the paper is preserved.

This applies also to the secondary outcomes. It is not OK to omit the between-group comparisons and confidence intervals, on the basis of a non-significant three-way comparison. Even though no equivalence limits were pre-specified (as far as I can see) for the secondary outcomes, as an equivalence study, the focus should remain on these treatment effect differences and their confidence intervals. Given the low event rates, these CIs will likely be wide, which limits what can be said about the equivalence, or otherwise, of the treatments, but that is to be expected for secondary outcomes.

Whether the width of the confidence intervals reported should have been adjusted for multiple comparisons is debateable, and given how close the confidence interval for the Arabin vs. Progesterone comparison comes to a 20% difference, this could perhaps be acknowledged in the discussion. I have to say that the justification for not applying an adjustment ("As all pairwise comparisons were of interest, a Bonferroni correction was not included") is not one I have heard before, and sounds wrong to me - surely the fact that all comparisons are of interest would support the need for adjustment?

Personally, I think that the equivalence limits of +/- 20% are quite wide. The treatment effect for all three treatments was assumed to be a 25% absolute reduction in PTB (from 75% to 50%). However, this was pre-specified, and accepted by the regulators, so cannot be changed. Nevertheless, whether a 20% treatment effect difference is sufficiently small to be considered equivalent could perhaps be discussed.

The primary analysis, as prespecified in the analysis plan, was "intention-to-treat". Given that ITT analyses tend to bias towards the null, equivalence studies would normally use a per-protocol population for the primary analysis. Saying that, it is usually recommended to look at the analysis in multiple ways, and the authors include a PP analysis, according to their analysis plan, which is good. However, what they report appears to be more of an "as-treated" analysis, rather than "per-protocol", since the denominator in the progesterone group is larger in Table 3 than in Table 2. For me, "per-protocol" would involve the exclusion of anyone who did not receive their allocated treatment, or otherwise deviated from the protocol. An "as-treated" analysis is fine to do, but should be specified as such.

Finally, there is a comment regarding the decision to end the study early which includes the text "the DMC (who reviewed the blinded data)". Usually, the DMC get to see unblinded data - is this a typo?

*** Reviewer #2: 

This randomised controlled trial randomised mixed high/ medium risk pregnancies with a short cervix to cerclage, progesterone 200mg od, or an Arabin pessary. Underpowered by its own, reasonable power calculation (halted due to COVID-19) it demonstrated no difference between the three treatments. However, more women receiving progesterone and an Arabin pessary subsequently developed bulging membranes and ultimately underwent rescue cerclage. It concludes that all three treatments are reasonable first line in this situation.

This was an ambitious multicentre trial, where all the right boxes were ticked. Women at this risk level are rare and difficult to recruit. The paper is well written and clear. 

The principal, obvious, problem, is its size and the ensuing the lack of statistical power. The design was for a primary outcome of birth <37 weeks; even if the numbers in the power calculation had been achieved, it would always have been under-powered for the more important secondary outcomes of birth <34 or 30 weeks, or severe morbidity. This was compounded by trial violations largely because of women's concerns. This issue needs to be better addressed in a limitations section, obvious though it is.

On design, the mixed risk groups make interpretation difficult. In real life, many women with a prior mid trimester birth or very preterm birth are started on progesterone, then serially scanned, and then have a cerclage if the cervix measures <25mm. Women with prior cervical surgery or an incidental finding of a short cervix do not receive progesterone until the cervix is found to be short. I do think the mix of 'risk' should be addressed in the limitations too.

Although randomisation was properly conducted, it is noteworthy that among women allocated cerclage, there was a higher (not statistically significant) number of women both with a second trimester birth and a very short cervix (<15mm). A woman with a prior history and cervix <15mm is very high risk; one with previous cervical surgery and a cervix >15mm and <25mm is not.

Table 1: the column headings are confusing: '% (n/128) or mean (SD)'. The formatting needs improvement.

Table 2: the 'Components of the adverse perinatal outcome' are incorrectly aligned. The formatting needs improvement.

The sentence '… we see ~40 in our large tertiary-referral preterm surveillance clinic per year…' seems out of place in a report of a multicentre trial. 

Despite its small size, this trial, which must have been very difficult to conduct, is clearly worthy of publication in a journal such as Plos Medicine.

*** Reviewer #3: 

Thank you for the opportunity to review this manuscript.

Please see my comments below:

Introduction

1. Short cervical length is typically considered less than or equal to 25mm

2. 'Mid-trimester' should be defined with a gestational age range

3. 'No single-study' - would avoid making claims unless including information regarding a systematic review with search terms etc. that support this statement

Methods

1. Should clearly state trial performed in singleton gestations only

2. Is short cervical length defined in this trial < 25mm or </=25mm?

3. What was the rationale for including gestational age as early as 14 weeks if trial interventions have been typically studied starting at 16 weeks in prior randomized trials— is cervical length screening starting at 14 weeks standard of care in the UK?

4. For patients with a prior late miscarriage after 16 weeks of gestation, is a history-indicated cerclage not be routinely performed? Or is this in the presence of preterm labor symptoms (i.e. pain or bleeding)? 

5. Regarding procedures for cerclage- if a patient had a short cervix at 23+6 for example, they may have had a cerclage placed up to 7 days later?

6. Did agents for tocolysis or antibiotics differ between centers?

7. Regarding procedures for vaginal progesterone- what was rationale for this dose and completion by 34 weeks instead of 36 weeks?

8. Why were cervical lengths performed after cerclage placement?

9. For the power analysis, none of the referenced studies had a PTB < 37 weeks rate as h

---

## [Decision Letter · Decision Letter 2]

23 May 2024

Dear Dr. Suff,

Thank you very much for re-submitting your manuscript "Comparing cervical cerclage, pessary and vaginal progesterone for prevention of preterm birth in women with a short cervix (SuPPoRT): A multi-centre randomised controlled trial" (PMEDICINE-D-23-03700R2) for consideration at PLOS Medicine.

I have discussed the paper with our academic editor and it was also seen again by three reviewers. I am pleased to tell you that, provided the remaining editorial and production issues are fully dealt with, we expect to be able to accept the paper for publication in the journal.

[LINK]

Please let me know if you have any questions, and we look forward to receiving the revised manuscript.   

Sincerely,

Richard Turner, PhD

Consulting Editor, PLOS Medicine

plosmedicine@plos.org

Requests from Editors:

Please amend the competing interest information (submission form) to include "AS is a member of PLOS Medicine's editorial board." or similar. 

Early in the abstract, we suggest amending the text to "... to our knowledge there has been no randomised ...".

Please add a few words to the abstract to quote aggregate demographic details for study participants. 

Early in the 'Methods and findings' subsection of the abstract, please state that this was an equivalence trial, and quote the equivalence margin.

Our academic editor requests that you quote pairwise comparisons for the primary outcome, with accompanying CI, in the abstract. 

At the end of the "Methods and findings" subsection of the abstract, please amend the relevant text in the abstract as follows: "A small proportion of women did not receive the intervention as per protocol, however per-protocol and as-treated analyses showed similar results. The main study limitation was that the trial was underpowered for neonatal outcomes, and was stopped early due to the COVID-19 pandemic."

In the abstract, immediately prior to the sentence stating the study limitations, please add a sentence, say, summarizing adverse events. 

Please relocate the 'Author summary' after the abstract. This should contain three subsections, each comprising three bulleted points, usually of a single sentence, each. You may find it helpful to consult a recently published paper in PLOS Medicine to get a sense of the preferred style. 

In the first paragraph of the Discussion (main text), please again amend the text to "... to our knowledge the only randomised trial".

Throughout the text, please style reference call-outs as follows: "... no benefit [9,10]." (noting the absence of spaces within the square brackets). 

In the reference list, please remove all italics. 

Where appropriate, 6 author names should be quoted, followed by "et al.".

Please use the journal name abbreviation "PLoS ONE".

Noting figure 1, please use the form "follow-up" throughout. 

Comments from Reviewers:

*** Reviewer #1: 

Alex McConnachie, Statistical Review

I thank the authors for their consideration of my original comments, and these are largely satisfactory.

My only quibble lies in in the fourth paragraph of the results. 

The sentence "There was no difference in the primary outcome (rate of PTB < 37 weeks) according to allocated treatment; cerclage 29.9% (38/127) vs. cervical pessary 31.1% (38/122) vs. progesterone 24.2% (32/132) overall difference p=0.4" is problematic, because a p-value of 0.4 does not mean there is "no difference" between the groups, simply that there is no evidence of a difference.

The next sentence refers the reader to table 2, since this shows "risk differences (p values) for individual intervention comparisons." For me, it is the confidence intervals in Table 2 (not the p-values) that matter. In order to say that there is "no difference" between the groups, in other words that the groups are equivalent, these confidence intervals need to lie within +/- 20% - the pre-defined equivalence limits - which they do. I would suggest that these risk differences and CIs should be reported in the fourth paragraph of the results, because for the primary outcome, these are the key results.

Otherwise, I have no further comments.

*** Reviewer #2:

The authors appear to have addressed all the comments made although I see some rebuff to the statistical review which I do not consider important.

This is good work and although the findings are not entirely conclusive it is now worthy of publication in such a journal as yours.

*** Reviewer #3: 

Thanks to the authors for their responses to my comments, which were adequately addressed.

However, I believe the manuscript should be updated to reflect the following:

Methods, comment 3: appropriate response, please update the manuscript to include cervical length screening was by provider discretion and not standard protocol

Methods, comment 7: appropriate response, please add reference to manuscript

Methods, comment 8: appropriate response, please provide rationale and add reference to manuscript

This manuscript will be a great contribution to the literature.

***

[LINK]

---

## [Editor Report · Decision Letter 3]

13 Jun 2024

Dear Dr Suff, 

On behalf of my colleagues and the Academic Editor, Dr Smith, I am pleased to inform you that we have agreed to publish your manuscript "Comparing cervical cerclage, pessary and vaginal progesterone for prevention of preterm birth in women with a short cervix (SuPPoRT): A multi-centre randomised controlled trial" (PMEDICINE-D-23-03700R3) in PLOS Medicine.

Prior to final acceptance, please also address the following points:

Please check that data are quoted consistently throughout the paper: it seems that "-0.7% [-12 to 10.7]" in the abstract and results section should be "-0.7% [-12.1 to 10.7]", looking at table 2.

Please quote "p<0.001" for small p values throughout, in place of the current "p=0.0003" in the results section, for example. 

Please expand the 'Author summary' to contain three subsections: 

"Why was this study done?

What did the researchers do and find?

What do these findings mean?"

Each subsection should consist of three bulleted points, ideally of 1-2 short sentences each. 

In the author summary, please use the active voice for preference (e.g., "We randomized ..." rather than "The researchers randomized ..."). 

PRESS

Sincerely, 

Richard Turner, PhD 

Consulting Editor, PLOS Medicine

plosmedicine@plos.org